# Primary Care Service Provision Scale for Evaluating the Right to Health Among International Migrant Populations

**DOI:** 10.3390/healthcare13162068

**Published:** 2025-08-21

**Authors:** Consuelo Cruz-Riveros, Alfonso Urzúa, Carolina Lagos, Evelyn Parada

**Affiliations:** 1Escuela de Enfermería, Universidad de las Américas, Concepción 3349001, Chile; 2Escuela de Psicología, Universidad Católica del Norte, Antofagasta 1270709, Chile; alurzua@ucn.cl; 3Escuela de Enfermería, Universidad Santo Tomás, Santiago 8370003, Chile; clagos17@santotomas.cl (C.L.); eve.parada01@gmail.com (E.P.)

**Keywords:** health, transients and migrants, health personnel, primary health care

## Abstract

**Introduction:** This study was conducted from July 2021 to December 2022. We propose a scale to measure the right to health among international migrants in primary care settings. The scale aims to highlight and objectively assess the elements integrated into the delivery of healthcare services by health personnel. **Objective:** Our aim was to develop and evaluate the psychometric properties of a measurement scale assessing the right to health in primary care for migrant populations in Chile, from the perspective of healthcare workers. **Methods:** An instrumental psychometric study was conducted. The sample comprised 339 primary healthcare workers from the Antofagasta, Biobío, and Metropolitan regions of Chile. The stages of the process included a theoretical review, conceptual definition, item construction, initial qualitative evaluation, and scale administration. **Results:** The initial 55-item model exhibited unsatisfactory fit indices (χ^2^ = 2608.693; df = 1271; *p* < 0.001; RMSEA = 0.056; CFI = 0.931; TLI = 0.919; SRMR = 0.054), whereas the refined 19-item model achieved satisfactory fit (χ^2^ = 441.72, df = 146, *p* < 0.001; RMSEA = 0.07; CFI = 0.95; TLI = 0.94; SRMR = 0.05). **Conclusions:** The scale demonstrates robust internal consistency and offers a valuable tool for evaluating primary healthcare delivery to international migrants based on the right to health framework.

## 1. Introduction

In 2020, an estimated 281 million people were living outside their country of birth [1]. Men accounted for the majority, 51.9%, a proportion that has risen steadily since 2000 as the share of female migrants has declined [1]. A central driver of these movements is people’s search for better living conditions compared to their countries of origin [2,3]. People migrate for employment, family reunification, or education, yet many are also compelled to move by armed conflict, violence, political crises, or natural disasters [1].

However, Latin America has not been immune to this phenomenon, with Chile ranking among the top five countries in the region with the largest international migrant populations, approximately 1.6 million people, or 8% of the national population, which increased exponentially during the pandemic [4].

Consequently, such population movements have prompted the development of policies and regulations at both global and local levels to ensure safe and timely responses to migrants’ actual needs, grounded primarily in the protection of human rights [2]. Among the social rights that elicit the greatest concern is the right to health; indeed, universal access serves as a critical component for safeguarding the health status of both adult and pediatric populations [5].

To anchor health policy decisions, the guidelines issued by the World Health Organization (WHO) establish the normative foundation for healthcare practices grounded in respect for every person’s fundamental right to health [2]. Accordingly, four inseparable dimensions constitute the right: availability, accessibility, acceptability, and quality and safety [2,6].

Despite these normative advances, progress in implementing policies that secure healthcare access has been uneven [7,8]. The chronic financial constraints of the health sector limit service capacity [7,8,9,10]. These system-level pressures are further compounded by additional factors that hinder the realization of the right to health, including systemic discrimination that disproportionately affects undocumented migrants, healthcare personnel’s beliefs and prejudices, standardized service protocols, lack of guidance on applying an intercultural approach in clinical practice, and the use of overly technical language that is difficult to understand [7,8,9,10].

Currently, there is no available instrument that encompasses all factors proposed by the WHO for assessing the realization of the right to health among migrant populations. Instead, fragmented indicators are used—such as coverage of services delivered to migrants or management-related measures focusing on user satisfaction and service quality [11].

Our objective is to develop and evaluate the psychometric properties of a scale that measures the right to health in primary care settings for migrant populations in Chile, from the perspective of healthcare workers. By identifying the variables that act as access barriers during service utilization, this tool can inform concrete actions to advance the full realization of migrants’ right to health.

## 2. Methods

We conducted an instrumental psychometric study [12,13].

### 2.1. Participants

A non-probability purposive sample of 339 healthcare workers was recruited between January and October 2022 from primary healthcare centers in the Antofagasta, Biobío, and Metropolitan regions of Chile. Inclusion criteria were age ≥ 18 years, an active employment contract at a primary healthcare center, and having attended at least one international migrant patient.

### 2.2. Procedures

The study was structured in five phases:Theoretical Review: Identification and analysis of factors related to the realization of the right to health;Conceptual Definition: Development of operational, semantic, and syntactic definitions within the right-to-health framework, resulting in four core dimensions (Table 1);Item Construction: Generation of questionnaire items, allocated across the four established dimensions;Initial Qualitative Evaluation: Conduct of cognitive interviews with primary healthcare personnel to assess item wording and comprehension. Participant feedback was used to modify and refine item phrasing;Field Administration and Quantitative Analysis: Criterion 1: Items displaying skewness or kurtosis > ±2 were removed because extreme values suggest ceiling/floor effects and limit discriminative power.

Item–Test Correlation; Criterion 2: Items with a corrected item–total correlation r < 0.30 were discarded for contributing insufficiently to the scale’s internal consistency.

Exploratory Factor Analysis (EFA); Criterion 3: EFA was used as an initial step to examine the latent structure of the instrument and to estimate the appropriate number of underlying factors. This stage aimed to identify the optimal number of dimensions representing the construct, based on criteria such as eigenvalues greater than one, the inflection point in the scree plot, and the proportion of explained variance. However, EFA results were not used as a basis for item elimination, since some overlapping among factors was theoretically expected due to the multidimensional nature of the construct and the correlation between factors.

Exploratory Structural Equation Modeling (ESEM): During the ESEM phase, items were removed based on three main criteria—(a) standardized factor loadings below 0.30; (b) cross-loadings on other factors equal to or greater than 0.40, which indicated a lack of discriminant validity; and (c) high residual correlations with other items, suggesting overlapping content or local misfit. Although the use of residual correlation as a criterion is more conservative, it was applied with caution and only when supported by additional evidence, such as low explained variance (R^2^) and weak theoretical alignment. This combined approach ensured the final model retained both empirical robustness and conceptual coherence.

Confirmatory Factor Analysis (CFA): Items that met the three previous filters were validated in an independent CFA to confirm the four-dimension structure and overall model fit.

Through these four sequential steps, the initial item pool was reduced to 19 final items, ensuring content validity, sound psychometric properties, and theoretical coherence with the WHO dimensions of the right to health.

### 2.3. Ethical Considerations

The project was approved by the Scientific Ethics Committee of the Universidad Católica del Norte (Resolution No. 015/2021). All participants provided written informed consent prior to their involvement in the study.

### 2.4. Instrument

The Primary Care Service Provision Scale for Evaluating the Right to Health among International Migrant Populations is a self-administered instrument designed for healthcare professionals. It assesses the four dimensions of right to health availability, accessibility, acceptability, and quality and safety, with items rated on a four-point Likert scale (1 = “strongly disagree,” 2 = “disagree,” 3 = “agree,” and 4 = “strongly agree”). Applying the scale helps identify specific aspects that can be improved throughout the care process for international migrant patients (see Table 1).

### 2.5. Data Analysis

#### 2.5.1. Descriptive and Psychometric Analysis

Descriptive statistics (mean, standard deviation, skewness, and kurtosis) were estimated using SPSSv25. Reliability analyses, including McDonald’s omega, Cronbach’s alpha, and item–dimension correlations, were conducted using JAMOVI version 1.8.4.0.

A preliminary exploratory factor analysis (EFA) was performed to examine the underlying structure and the relationships between items and factors. Subsequently, an Exploratory Structural Equation Modeling (ESEM) analysis was carried out in Mplus 8, assessing factor loadings and model fit indices.

#### 2.5.2. Construct Validity Analysis

Confirmatory factor analysis was conducted in Mplus using the weighted least squares means and variance adjusted (WLSMV) estimator, which is robust to violations of normality. Model fit was evaluated using conventional criteria: Comparative Fit Index (CFI) and Tucker–Lewis Index (TLI) > 0.90; Root Mean Square Error of Approximation (RMSEA) between 0.05 and 0.08; and Standardized Root Mean Square Residual (SRMR) < 0.08 [15,16,17,18].

## 3. Results

The mean participant age was 34.8 years (SD = 9.3), 239 (70.5%) participants were female, and the mean duration of professional experience was 8.6 years (SD = 7.8). The most common occupation was higher-level nursing technicians (TENS), representing the largest group (n = 122; 36%), followed by nurses (n = 60; 17.7%), administrative staff (n = 28; 8.3%), kinesiologists (n = 22; 6.5%), and physicians (n = 19; 5.6%).

### 3.1. Preliminary Item Analysis

A preliminary descriptive analysis of the items was conducted to identify those that failed to meet the selection criteria related to distribution (skewness and kurtosis) and to the item–total correlation. Items that did not reach these thresholds were excluded from subsequent factor analyses.

Descriptive statistics were calculated for all 56 items (Appendix A). As shown, the highest mean scores were observed for items “ac1” (M = 3.55) and “acep13” (M = 3.56). Regarding variability, the items “ac4” (SD = 1.01) and “disp3” (SD = 1.03) exhibited the greatest dispersion, as indicated by their standard deviations. Item Distribution Analysis upon examining the distribution of participants’ responses, most skewness and kurtosis values were negative, as detailed in Appendix A. According to the predefined criteria, both skewness and kurtosis values greater than +2 or less than −2 were considered unacceptable, leading to the exclusion of items that did not meet this requirement (Appendix A) [17,18]. A second criterion for item selection was the item–test correlation, with a threshold of r > 0.30; consequently, item ac3 was excluded [17,18,19]. Furthermore, the overall internal consistency of the scale was assessed using Cronbach’s alpha (α = 0.96) and McDonald’s omega (ω = 0.96), both indicating excellent reliability, as they exceed commonly accepted thresholds [20]. However, the internal consistency for the accessibility dimension was lower, with Cronbach’s α = 0.78 and McDonald’s ω = 0.81, although both values remain within acceptable limits (Table 2).

### 3.2. Exploratory Structural Equation Modeling

As a preliminary step to the ESEM analysis, an exploratory factor analysis (EFA) was conducted to examine the relationships between items and underlying factors. Sampling adequacy was assessed using the Kaiser–Meyer–Olkin (KMO) index, which yielded a global value of 0.50, considered the minimum acceptable threshold for factor analysis [18]. Bartlett’s test of sphericity was significant (χ^2^ = 45,457.644, df = 1485, *p* < 0.001), indicating that the correlations among items were sufficient to justify factor analysis. Although eigenvalues greater than 1 suggested up to nine factors, the final solution was determined based on theoretical and psychometric criteria, resulting in a four-factor model that explains 44% of the total variance (Table 3).

An exploratory analysis was initiated using Exploratory Structural Equation Modeling (ESEM) to examine whether the items clustered into the four theoretical dimensions of the right to health. The analysis began with 55 items, considering those with standardized factor loadings below 0.30 for elimination, as this was established as the minimum acceptable threshold.

The initial model with 55 items showed an acceptable fit (χ^2^ = 2608.693; df = 1271; *p* < 0.001; RMSEA = 0.056; CFI = 0.931; TLI = 0.919; SRMR = 0.054). Item removal was guided by several criteria: (1) standardized factor loadings below 0.30 [18,21]; (2) cross-loadings ≥ 0.30 on unintended factors, which suggested conceptual overlap [22]; and (3) substantial residual correlations between items, which indicated potential local misfit. Based on these criteria, the following items were removed: disp1, disp2, disp3, disp4, disp5, disp6, disp7, disp10, disp12, disp13, disp15, disp17, ac1, ac2, ac4, ac6, ac8, ac9, ac11, ac13,acep2, acep3, acep4, acep6, acep7, acep8, acep11, acep12, acep13, acep14, acep15, cal1,cal2, cal3, cal6 and cal8. A detailed summary of item-level decisions and statistical indicators is available in Appendix A.

Subsequently, the adjusted model with 19 items demonstrated good overall fit. The fit indices indicated that the model was adequate: the RMSEA fell within the acceptable range (RMSEA < 0.08), while both the CFI (0.97) and TLI (0.96) exceeded the recommended threshold of 0.90. Additionally, the SRMR value was low (0.03), further supporting the model’s acceptable fit to the data (Table 4).

### 3.3. Confirmatory Factor Analysis

Confirmatory factor analysis (CFA) was conducted to evaluate the final 19-item structure identified in the previous exploratory model (ESEM). The four-point ordinal data were analyzed using the WLSMV estimator [23,24]. The confirmatory model demonstrated good fit (χ^2^ = 441.724, df = 146, *p* < 0.001; RMSEA = 0.07; CFI = 0.95; TLI = 0.94; SRMR = 0.05; Table 4). Although the RMSEA value is within the acceptable range, it exceeds the optimal threshold (<0.05). Nevertheless, the four-factor solution provided an adequate representation of the data. Internal consistency was excellent, with Cronbach’s α = 0.90 and McDonald’s ω = 0.91, indicating reliable measurement of the construct (Table 5).

The items showing high cross-loadings, such as acep1, ac5, and ac12, were retained in the model, as they theoretically reflect transversal aspects that result in conceptual overlap among the different dimensions. This overlap is consistent with the complex and interconnected nature of the components involved in the right to healthcare (Table 5).

## 4. Discussion

### 4.1. Main Findings

This study aimed to present the development and psychometric evaluation of a scale measuring the right to health in primary care for migrant populations, from the perspective of healthcare personnel. This provides preliminary empirical evidence of the instrument’s validity. The developed questionnaire, composed of 19 items, assesses the four dimensions that constitute the right to health according to the guidelines of the World Health Organization (WHO) [14]. The final factorial structure revealed significant correlations among the factors, reflecting the transversal nature of the elements assessed throughout the healthcare process for international migrants. Internal consistency was evaluated using both Cronbach’s alpha and McDonald’s omega coefficients, acknowledging that relying solely on alpha may overestimate reliability when items have unequal loadings or when the scale includes a small number of items [25]. The inclusion of both indices offers a more accurate and robust estimate of the instrument’s reliability.

### 4.2. Comparison with Related Instruments

Although various instruments exist, the only one that explicitly addresses the right to health, albeit from the user’s perspective, is the Experience in Exercising the Right to Health Care Scale (EERHC) [26]. Other alternatives do not evaluate the right as the central focus, but rather specific components, such as patient satisfaction (SERVQUAL [27]) or staff attitudes toward immigrant patients (AHPI) [28]. Internationally, widely used tools such as the Primary Care Assessment Tool (PCAT) and the WHO Health System Responsiveness measures focus on primary care attributes and user experience, or on system surveillance through indicators of effective coverage and financial protection [28,29,30]. In the same vein, the literature organizes metrics into structural, process, and outcome indicators to assess access and service management [31], along with population health outcomes (birth and mortality rates, prevalence, and incidence) [32,33].

The difference from previous proposals lies in the fact that our scale explicitly operationalizes the WHO right-to-health framework (availability, accessibility, acceptability, and quality) from the perspective of healthcare personnel, specifically through 19 items with confirmatory psychometric evidence for a four-factor model. This proposal, particularly pertinent for Latin American contexts characterized by South–South migration, enables monitoring and cross-context comparison, and due to its brevity and operational language, it has the potential to be used by non-governmental organizations (NGOs) and community providers. Moreover, it is amenable to translation and cross-cultural validation. The results obtained with this instrument can contribute to the monitoring and follow-up of public policies established in governmental regulations that promote a rights-based approach to health [33,34,35]. Nevertheless, the literature shows significant gaps between the theoretical design of these policies and their effective implementation in primary care [36,37]. In this context, tools such as the proposed scale enable the timely identification of risk factors across the four dimensions of the right to health, supporting the maintenance or improvement of health outcomes in migrant populations [14,33,34,35]. Even so, structural challenges persist, translating into standardized care practices, which, in many cases, do not adequately consider cultural and linguistic relevance [33,34,35,36,37,38].

### 4.3. Limitations

This study has several limitations. First, it was based on non-probability sampling and involved a relatively small group of healthcare professionals, both clinical and administrative. Second, the self-report format may be subject to recall and social desirability biases. Third, the scale captures staff perceptions at a single point in time, and such perceptions may not remain stable over time. Fourth, there is a need to analyze how the scale performs across different professional roles and geographic areas. In addition, the sample included only personnel from urban primary care centers. These limitations highlight the need for future research with larger, probability-based samples, longitudinal designs, and triangulation with patient perspectives.

Future research should include systematic translation and cultural adaptation and test measurement invariance across cultural groups and professional strata.

## 5. Conclusions

In summary, our scale, developed to measure the right to health in primary care for migrant populations based on healthcare workers’ perceptions, comprises nineteen items across four dimensions (availability, accessibility, acceptability, and quality–safety). Psychometric testing confirms a robust internal structure and supports its usefulness, within the right-to-health framework, as a tool for evaluating primary care service delivery for international migrants. Future studies should enlarge the sample and include more diverse geographical settings, since the instrument enables self-assessment of the components needed to ensure care that is fully consistent with a rights-based approach.

## Figures and Tables

**Table 1 healthcare-13-02068-t001:** Number of items in the initial version of the Primary Care Service Provision Scale for Evaluating the Right to Health among International Migrant Populations [14].

Dimension	Definition	Items
Availability	Understood as the “guarantee of general supplies, including facilities, goods, and services” [14]	17
Accessibility	Conceptualized as the “principle of non-discrimination—covering the availability component, governance, geographic location, and hours of operation—such that no group is disadvantaged based on age, gender, or legal status in the country when accessing promotion, prevention, curative, and rehabilitation services” [14]	13
Acceptability	Defined as “actions focused on gender, cultural appropriateness, and respect for medical ethics, incorporating measures to reduce linguistic and cultural barriers, including cultural mediation, interpretation, and translation” [14]	15
Quality and Safety	Refers to “ensuring the established minimum standard, thereby enabling the safe provision of care within the domains covered by availability” [14]	11

**Table 2 healthcare-13-02068-t002:** Reliability indices by dimension of the healthcare service delivery scale.

	Dimension		Scale	
	Cronbach’s	McDonald’s	Cronbach’s	McDonald’s
	α	ω	α	ω
Availability (Disp)	0.9	0.9		
Accessibility (Ac)	0.78	0.81	0.96	0.96
Acceptability (Acep)	0.9	0.9		
Quality and Safety (Cal)	0.9	0.91		

**Table 3 healthcare-13-02068-t003:** Characteristics of the factors.

		Unrotated Solution			Rotated Solution		
	Eigenvalues	Sum of Squared Loadings	Proportion Var.	Cumulative	Sum of Squared Loadings	Proportion Var.	Cumulative
Factor 1	21.19	20.81	0.37	0.37	9.59	0.17	0.17
Factor 2	5.51	5.14	0.09	0.47	5.94	0.10	0.28
Factor 3	3.29	2.89	0.05	0.52	4.71	0.08	0.36
Factor 4	2.23	1.86	0.03	0.55	3.93	0.07	0.44
Factor 5	2.13	1.72	0.03	0.59	3.9	0.07	0.51
Factor 6	1.64	1.22	0.02	0.61	1.85	0.03	0.54
Factor 7	1.45	1.00	0.01	0.63	1.54	0.02	0.57
Factor 8	1.28	0.88	0.01	0.64	1.01	0.01	0.59
Factor 9	1.15	0.76	0.01	0.66	0.65	0.01	0.60

**Table 4 healthcare-13-02068-t004:** Summary of model fit indices.

Model		χ^2^	df	*p*-Value	RMSEA (IC)	CFI
ESEM	5573.98	171	0.05 (0.04–0.06)	0.98	0.97	0.03
CFA	441.72	146	0.07 (0.06–0.08)	0.95	0.94	0.05

**Table 5 healthcare-13-02068-t005:** Factor loadings and correlations of the evaluated measurement models.

	ESEM				CFA			
Items	F1	F2	F3	F4	F1	F2	F3	F4
Disp8	**0.89**	−0.17	0.14	−0.06	**0.72**			
Disp9	**0.87**	−0.03	0.03	−0.14	**0.68**			
Disp11	**0.5**	0.28	−0.29	0.15	**0.69**			
Disp14	**0.48**	0.13	−0.02	0.03	**0.65**			
Disp16	**0.43**	0.3	−0.19	0.23	**0.78**			
Acep1	0.39	**0.4**	0.11	0.1		**0.76**		
Acep5	0.2	**0.44**	−0.09	0.06		**0.51**		
Acep9	0.11	**0.48**	0.3	0.14		**0.82**		
Acep10	0.08	**0.42**	0.36	0.17		**0.78**		
Ac5	0.16	0.28	**−0.3**	0.11			**0.31**	
Ac7	0	0.37	**0.56**	0.02			**0.66**	
Ac10	0	0.36	**0.42**	0.09			**0.66**	
Ac12	0.08	−0.05	**0.3**	0.6			**0.86**	
Cal4	0.18	0.08	0.27	**0.45**				**0.78**
Cal5	−0.07	0.11	−0.03	**0.8**				**0.79**
Cal7	0.15	−0.16	0.07	**0.84**				**0.85**
Cal9	−0.09	0.17	−0.06	**0.75**				**0.75**
Cal10	−0.05	0.1	0.02	**0.83**				**0.86**
Cal11	0.12	−0.07	0.15	**0.81**				**0.91**
F1	1				1			
F2	0.44	1			0.73	1		
F3	0.3	0.2	1		0.6	0.8	1	
F4	0.48	0.53	0.42	1	0.6	0.79	0.8	1

F1: availability (Disp); F2: acceptability (Acep); F3: accessibility (Ac); F4: quality and safety (Cal).

## Data Availability

The data presented in this study are available on request from the corresponding author.

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
