# Peer review of "Primary Care Service Provision Scale for Evaluating the Right to Health Among International Migrant Populations"

_healthcare, 2025, doi:10.3390/healthcare13162068_

Round 1
Reviewer 1 Report
Comments and Suggestions for Authors
- How this affects the psychometric validity of the scale across broader populations???
- You included 221 participants where is formal justification for sample size adequacy in relation to factor analysis??? For EFA/CFA, a commonly used rule is 5-10 participants per item. Please clarify whether the final sample meets this criterion especially for the initial 56-item version?
- The decision to eliminate items based on factor loadings < 0.40, item-test correlations < 0.30, and skewness/kurtosis > ±1.5 is briefly mentioned but not supported by figures or specific cutoffs in all cases. Please provide a clearer justification and a summary table of excluded items with their psychometric properties.
- Cronbach’s alpha is reported that’s good, it assumes tau-equivalence and may overestimate reliability when item correlations are high. Since McDonald’s w is also available, the emphasis should shift to “w” as the more appropriate reliability measure for multidimensional constructs.
- The RMSEA = 0.06 for both ESEM and CFA models is on the border of acceptable fit. please include confidence intervals for RMSEA to interpret model fit more accurately.
- Confirmatory Factor Analysis (CFA) - Cross-Loadings Not Shown
- No test for measurement invariance (by gender, profession, or region) is reported. This is crucial for scale validation, especially in culturally and demographically diverse samples. Consider adding or discussing this limitation.
- The Likert items are correctly modeled using WLSMV estimator, no mention is made of whether ordinal-to-continuous assumptions were evaluated.
- Include a scree plot or eigenvalue table to justify the number of factors retained.
- Present item-total correlation ranges by subscale to show consistency.
- provide model modification indices or residual plots to assess local misfit.
Best of Luck
Author Response
"Please see the attached file."

Reviewer 2 Report
Comments and Suggestions for Authors
Thank you for the opportunity to review this article. Please find my suggestions.
ABSTRACT
-Please, include the date of the study in “Abstract” section.
MATERIAL AND METHODS
-Lines 67-70: “The mean participant age was 34.3 years (SD = 9.1), 156 (70.1 %) were female, and the mean duration of professional experience was 8.4 years (SD = 7.7). The most common occupation was higher-level nursing technicians (TENS; n = 68, 30.7 %).” Please, this information could be moved to “Results” section. (Sociodemographic information of the participants).
RESULTS
-The authors described only 30 items. Could the authors include all 56 items as an appendix?
-Line 18: “The initial 56-item model…” / Line 128: “The initial analysis of the 54 items using…” 54 or 56? Please, correct the information.
CONCLUSION
Lines 176-178: “The Right to Health measurement scale in primary care for migrant populations comprises thirty items across four dimensions: availability, accessibility, acceptability, and quality and safety.” OR “In this study the Right to Health measurement scale in primary care for migrant populations comprises thirty items across four dimensions: availability, accessibility, acceptability, and quality and safety.” Please, think about this.
Author Response
"Please see the attachment."

Reviewer 3 Report
Comments and Suggestions for Authors
I would like to recommend the following changes to improve the manuscript.
An introduction that is more organised and targeted would be beneficial.
Currently, some paragraphs combine context, argumentation, and supporting details in ways that obscure their meaning. I suggest dividing the section into four parts:
(1) the historical context of global migration;
(2) the current situation in Chile and Latin America;
(3) the health rights concerns of migrants; and
(4) the aim of the study. Less repetition is also necessary (for instance, the phrase "safe and timely responses" appears in both the abstract and the introduction).
The materials and methods section need to be significantly revised. The methodology is generally well described, though there are a few points that require clarification. A concise definition of the instrument is required for a broader audience. Although the five-phase process is clearly outlined, the criteria for item reduction and refinement - specifically, the reduction from 56 to 30 items - could be more transparent.
It is also necessary to define or cite the "semantic and syntactic definitions" and other terms listed in Table 1. From a formatting perspective, some punctuation mistakes need to be fixed. Despite being thorough, the results section contains some very technical information. For example, because of its high density, Table 2 may be more appropriate as supplemental material.
More detailed explanations of the reasoning behind the selection of these specific models and how they fit into the theoretical framework of the scale would be beneficial, even though ESEM and CFA are covered in detail without any interpretive context. Statistics back up the validity and reliability claims, but more interpretive justification would be helpful.
I recommend dividing the results and discussion sections into chapters to facilitate comprehension. The discussion currently restates findings without offering a more comprehensive theoretical background or interpretation. For instance, the lower reliability of some subscales (such as accessibility) is mentioned but not discussed. Because there is little reference to earlier instruments or research on evaluating healthcare access for migrants, it is also difficult to evaluate the study's contribution in context.
The limitations section is a bit brief; additional information about potential sample bias and the need for cross-cultural validation would improve the discussion. More research articles from the last ten years should be included, in my opinion.
The conclusion was revised and expanded. I recommend describing how this scale might be applied in practice as well as the next steps in its implementation or validation. Including recommendations for more research would also increase the section's impact.
Author Response
"Please see the attached file."

Round 2
Reviewer 3 Report
Comments and Suggestions for Authors
Please add in discussion a brief comparison with similar instruments used in other countries or healthcare contexts.Under limitations and future research, include:
1. Potential for translation and validation in other languages/cultures.
2. Possible use of the instrument by NGOs and community-based providers.
Also, consider structuring the discussion into clear sub-sections.
Author Response
"Please see the attachment."
